# Veteran and first responder family members show distinct mental health networks centered on negative emotions
Johannes A. Karl[1,2], Warren N. Ponder [3], Jose Carbajal[4] & Oleg N. Medvedev [5] ✉

The interplay of mental health symptoms among family members of veterans and first responders remains poorly understood despite their vital support role. Network analysis and community detection were performed on mental health assessment data from 317 treatment-seeking family members of trauma-exposed veterans and first responders, who completed clinical distress measures including posttraumatic stress disorder, depression, and anxiety. Network analysis revealed six distinct symptom communities: depression, generalized anxiety disorder, intrusion and avoidance, anxious arousal, externalizing behaviors, and negative alterations. Strong negative feelings (fear, horror, anger) and uncontrollable worry emerged as the most influential nodes in the network. Remarkably, 55.5% of participants screened positive for probable posttraumatic stress disorder, while 38.5% reported moderately severe to severe depression, and 36.6% experienced severe generalized anxiety disorder. The network demonstrated high stability across bootstrap analyses, with a correlation stability coefficient exceeding 0.59. Overall, this study revealed network of co-occurring mental health symptoms in family members of veterans and first responders. The identification of six distinct symptom communities suggests that traditional diagnostic boundaries may not fully capture the complexity of psychological distress in this population. These findings highlight the need for targeted interventions addressing both fear-based trauma symptoms and mood dysregulation in this understudied group.

Combat exposure negatively impacts the mental health of deployed military personnel[1]. Post-9/11 combat veterans show posttraumatic stress disorder prevalence estimates ranging from 14 to 16%[2] to as high as 65–70%[3,4], with substantial comorbidity including depression (43%–48%) and anxiety disorders (12%–13%)[5,6]. Similarly, first responders—including law enforcement officers, emergency medical technicians, and firefighters—face elevated trauma exposure[7,8] and mental health challenges, with posttraumatic stress disorder rates reaching 22% among law enforcement officers and 51% among firefighters and emergency medical technicians[9].

The psychological impact of trauma exposure extends beyond the individual to affect entire family systems[10]. Family members of trauma-exposed personnel experience secondary stressors through witnessing their loved one's symptoms, managing household disruptions, and adapting to behavioral changes. This secondary trauma exposure manifests through multiple pathways: emotional contagion from direct symptom observation, systemic family disruption affecting communication patterns and role distributions, and chronic stress from caregiving responsibilities. Military spouses show comparable rates of depression to the general population (4.9%)[11], yet face unique stressors including deployment cycles and military culture demands[12]. Among caregivers of veterans with spinal cord injuries, significantly higher depression and lower quality of life emerge compared to general populations[13].

These family impacts can perpetuate across relationships and potentially across generations[14]. Studies of veteran families reveal patterns of strained parent-child relationships[15], emotional dysregulation in offspring[16], and communication difficulties around mental health topics[17]. Children in veteran households show 1.45 times greater risk of mental illness compared to non-veteran households, with particular increases in externalizing behaviors[18]. First responder families similarly experience elevated psychological distress, with 82% of emergency dispatcher family members observing work-related stress transmission[19] and children showing increased posttraumatic stress disorder, depression, and anxiety symptoms[20,21].

[1]Department of Psychology, University of Zurich, Zurich, Switzerland. [2]Victoria University of Wellington, Wellington, New Zealand. [3]One Tribe Foundation, Euless, TX, USA. [4]Stephen F. Austin State University, Nacogdoches, TX, USA. [5]School of Psychological and Social Sciences, University of Waikato, Hamilton, New Zealand. ✉e-mail: oleg.medvedev@waikato.ac.nz

Understanding the co-occurrence and clustering of mental health symptoms in these family members requires sophisticated analytical approaches. The quadripartite model proposes that posttraumatic stress disorder, depression, and generalized anxiety disorders share underlying dimensions of general distress and symptom specificity[22]. Network analysis offers a complementary framework by modeling symptoms as inter-connected nodes[23], revealing central symptoms that may serve as inter-vention targets[24] and identifying symptom communities that transcend traditional diagnostic boundaries. Community detection within networks can uncover empirical symptom clusters, providing insights into how psychological distress organizes naturally rather than according to pre-determined diagnostic categories[25].

Previous network analyses of veterans[26], first responders[27], and trauma-exposed civilians[28] have identified distinct symptom communities sup-porting the quadripartite model, yet no studies have examined these patterns in family members. This population represents a unique intersection of secondary trauma exposure and caregiving stress that may produce distinct symptom presentations. We conducted network analysis and community detection on mental health assessment data from 317 treatment-seeking family members of trauma-exposed veterans and first responders. We anticipated identifying symptom communities including depression, generalized anxiety, intrusion and avoidance, anxious arousal, and negative alterations, while remaining open to discovering additional clusters specific to this population's secondary trauma and caregiving experiences.

## Methods
### Participants
The sample consisted of 317 treatment-seeking family members of trauma-exposed first responders and/or veterans. The average age was 38.28 years ($SD = 12.44$), mostly female 82.0% ($n = 260$), and white 71.3% ($n = 226$). Gender was ascertained through self-report during intake assessment. The respondents were either immediate and/or extended family relationships (i.e., sibling, aunt, uncle, cousin, grandparent). Most were significant others 57.1% ($n = 181$) and 65.7% ($n = 208$) completed "some college" or higher. Of the individual participants, all of whom were over 18 years old, 58.7% ($n = 186$) reported being married. In this sample, 65.3% ($n = 207$) were family of veterans and 34.7% ($n = 110$) were family of first responders. Race and ethnicity data were collected through self-report and are presented in Table 1. Participants received no compensation for this research as data were collected during routine clinical intake.

### Procedure
Data were collected from family members who sought counseling services between 2015 and 2021 at a non-profit organization serving veterans, first responders, frontline healthcare workers, and their families. Demographic data and standardized assessments were completed at their initial appointment before the client began treatment. Inclusion criteria were being related to a first responder or veteran, over the age of 18, and completing baseline assessments that were verified for completeness by the intake manager. No participants meeting these criteria were excluded from analyses. This study was not pre-registered due to its exploratory nature.

### Ethics statement
This study was approved by Stephen F. Austin State University Institutional Review Board, ensuring compliance with ethical standards for research involving human participants. All participants provided informed consent, and the study adhered to principles of inclusivity, transparency, and responsible data collection.

### Measures
The PTSD Checklist-5 (PCL-5) is a 20-item measure that assesses post-traumatic stress disorder[29]. Individual responses range from 0 (not at all) to 4 (extremely) and the aggregated score ranges from 0 to 80. Higher scores indicate more severe posttraumatic stress disorder symptomatology. PCL-5

**Table 1 | Demographics**

| Characteristic | ($n = 317$) |
|---|---|
| Age (Years) | |
| Mean | 38.28 |
| Median | 36.00 |
| D | 12.44 |
| Range | 54 |
| Individual relationship status | |
| Single | 38 (12.0%) |
| Committed relationship | 44 (13.9%) |
| Married | 186 (58.7%) |
| Separated | 13 (4.1%) |
| Divorced | 13 (4.1%) |
| Remarried | 1 (0.3%) |
| Widowed | 14 (4.4%) |
| Missing | 8 (2.5%) |
| Familial relationship status | |
| Family | 129 (40.7%) |
| Adult children | 7 (2.2%) |
| Significant other | 181 (57.1%) |
| Relationship with Veteran/First Responder $n$ (%) | |
| Veteran | 207 (65.3%) |
| First responder | 110 (34.7%) |
| Education $n$ (%) | |
| Below high school | 9 (2.7%) |
| High school/GED | 100 (31.6%) |
| Some college | 61 (19.3%) |
| Associates degree | 35 (11.1%) |
| Bachelors degree | 64 (20.2%) |
| Graduate | 26 (8.2%) |
| Missing | 22 (6.9%) |
| Gender $n$ (%) | |
| Women | 260 (82.0%) |
| Men | 56 (17.7%) |
| Non-binary | 1 (0.3%) |
| Ethnicity $n$ (%) | |
| African American/Black | 29 (9.1%) |
| Asian American | 8 (2.5%) |
| Latino(a)/Hispanic | 39 (12.4%) |
| Multiple ethnicities | 8 (2.5%) |
| Native American | 5 (1.6%) |
| White | 226 (71.3%) |
| Hawaiian/Pacific Islander | 2 (0.6%) |

scores 33 or higher indicate a probable posttraumatic stress disorder diagnosis[30]. In this sample, the Cronbach's alpha was $\alpha = 0.94$.

The Patient Health Questionnaire-9 (PHQ-9) is a 9-item assessment that assesses depression[31]. Individual responses range from 0 (not at all) to 3 (nearly every day) and the summed score ranges from 0 to 27. Higher scores indicate more severe depression. Aggregated scores from 0 to 4 is minimal, 5 to 9 mild, 10 to 14 moderate, 15 to 19 moderately severe, and 20 to 27 is considered severe depression[31]. In this sample, the Cronbach's alpha was $\alpha = 0.88$.

The Generalized Anxiety Disorder-7 (GAD-7) is a 7-item scale that assesses generalized anxiety[32]. Individual responses range from 0 (not at all)

to 3 (nearly every day) and the summed score ranges from 0 to 21. Higher scores indicate more severe generalized anxiety. Aggregated scores from 0 to 4 is minimal, 5 to 9 mild, 10 to 14 moderate, and 15 to 21 is considered severe generalized anxiety[32]. In this sample, the Cronbach's alpha was $\alpha = 0.90$.

## Data analyses

The data in this study were screened and found to be normally distributed. We conducted a two-sided independent samples $t$-test comparing family of veterans and first responders. The network model estimated in this study was computed using R (version 4.4.0). The Gaussian graphical model network approach was applied to model partial correlations[33] among nodes, representing item-level symptoms of posttraumatic stress disorder, depression, and generalized anxiety. Consistent with previous research on co-occurring psychiatric symptoms[28], overlapping symptoms were excluded: PHQ-9 item 3 (sleep difficulties, redundant with PCL-5 item 20, $r = 0.69$), GAD-7 item 5 (restlessness, redundant with PHQ-9 item 8, $r = 0.51$), PCL-5 item 12 (loss of interest in pleasurable activities, redundant with PHQ-9 item 1, $r = 0.60$), and PCL-5 item 19 (concentration difficulties, redundant with PHQ-9 item 7, $r = 0.66$). As a result, each symptom was represented only once in the network.

Network associations (i.e., edge-weights) were regularized using the Extended Bayesian Information Criterion with the Graphical Least Absolute Shrinkage and Selection Operator. Through iterative regularization, this method retrieves the optimal network, increasing parsimony by eliminating spurious edges[34]. The qgraph package[35] was used to visualize the network, with a multidimensional scaling-based layout to ensure interpretable locations and distances for nodes. Green lines between nodes represent positive edge-weights, while red lines indicate negative edge-weights. The thickness and translucency of the lines correspond to the size of the edge-weights, with greater absolute edge-weights appearing thicker and darker.

The reliability of the edge-weights was assessed using 1000 bootstrapped samples with the bootnet package[33]. The overall edge stability of the case-bootstrapped network was evaluated using the correlation stability coefficient, which is usually compared to the commonly accepted threshold for acceptable stability of 0.59[33]. A metric for node centrality (i.e., expected influence) was then computed from the network structure for each individual node. One-step expected influence is the sum of all direct edge-weights from one node to all other nodes[24]. A node with high expected influence suggests it holds an impactful position in the network. Community detection analysis was completed using the Louvain algorithm from the Exploratory Graph Analysis package[25], which estimates a Gaussian graphical model network to identify clusters of nodes. This approach has shown improved community identification precision compared to methods such as exploratory factor analysis. The Louvain algorithm identifies communities within a network by optimizing modularity, a measure that quantifies the strength of division of a network into clusters. Nodes within the same community have denser connections with each other than with nodes in different communities. The algorithm works through an iterative process, beginning with each node as its own community, then progressively merging communities to maximize the modularity score. This optimization occurs by local improvements, where nodes are moved between communities if such movement increases the overall modularity, followed by aggregation, where the identified communities are collapsed into supernodes. These steps repeat until no further improvement in modularity is possible. To assess the stability of the extracted solution, we bootstrapped the solution using the Exploratory Graph Analysis boot function, which generates multiple resamples of the original data and applies the community detection algorithm to each resample[36]. This procedure allowed us to evaluate consistency in the number of communities detected, stability of item assignments to communities, and overall robustness of the community structure.

## Reporting summary

Further information on research design is available in the Nature Portfolio Reporting Summary linked to this article.

## Results

### Descriptive statistics

The average PCL-5 score was 34.51 ($SD = 18.84$), PHQ-9 score was 12.27 ($SD = 6.68$), and GAD-7 score was 11.94 ($SD = 5.72$). Using a PCL-5 cutoff score of 33 or higher, 55.5% of participants met criteria for probable posttraumatic stress disorder. For depression severity, 14.1% had minimal, 23.7% mild, 23.7% moderate, 22.1% moderately severe, and 16.4% severe depression. For generalized anxiety severity, 10.7% had minimal, 24.6% mild, 28.1% moderate, and 36.6% severe generalized anxiety. There were no statistically significant differences between PCL-5 scores for the family of veterans (34.62; $SD = 19.46$) and the family of first responder (34.32; $SD = 17.69$), $t(315) = 0.14$, $p = 0.89$, $d = 0.02$, 95% CI [−0.22, 0.25]; between PHQ-9 scores for the family of veterans (12.35; $SD = 6.82$) and the family of first responder (12.13; $SD = 6.45$), $t(315) = 0.29$, $p = 0.78$, $d = 0.03$, 95% CI [−0.20, 0.27]; and between GAD-7 scores for the family of veterans (11.87; $SD = 5.97$) and the family of first responder (12.08; $SD = 5.26$), $t(315) = −0.32$, $p = 0.75$, $d = −0.04$, 95% CI [−0.27, 0.20].

### Network parameters

Based on statistical power analysis, our sample size of 317 participants exceeded the minimum required sample size of 311 needed to estimate the empirical network with a power of 0.80 and sensitivity of 0.60, as determined through bootstrap simulations (Supplementary Fig. S1). This ensures adequate statistical power for reliable network inference while maintaining acceptable false positive rates in our analysis.

Network estimation identified 32 nodes representing mental health symptoms across the three measures. The stability analysis yielded a correlation stability coefficient of $r = 0.70$ (Fig. 1), which was above the commonly established threshold of 0.59. Bootstrap stability analysis across different sample sizes showed most network metrics achieved stability above the 0.70. Network centrality analysis revealed PCL-11 (Having strong negative feelings such as fear, horror, anger, guilt, or shame) and GAD-2 (Not being able to stop or control worrying) as the nodes with highest expected influence in the network. As shown in Table 2, PCL-11 demonstrated the highest Expected Influence-1 (1.17) and Expected Influence-2 (2.32), followed by GAD-2 with Expected Influence-1 (1.12) and Expected Influence-2 (2.22). Expected influence values across all nodes ranged from 0.54 (PCL-8) to 1.17 (PCL-11). Network edge analysis revealed large

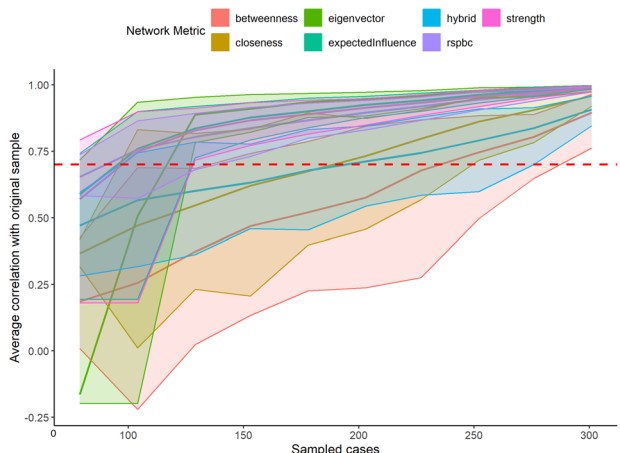

**Fig. 1 | Stability of network characteristics across case-resamples.** Lines show the correlation between original network characteristics and characteristics of networks computed from bootstrapped samples as a function of sample size ($n = 317$). Different colored lines represent different network centrality metrics: betweenness (blue), closeness (green), eigenvector (red), expected influence (purple), hybrid (orange), and strength (cyan). Shaded areas represent 95% confidence intervals around each metric. The horizontal red dashed line at 0.70 indicates the recommended stability threshold for adequate network reliability. Most metrics demonstrate adequate stability with correlation coefficients above the threshold.

## Table 2 | Node-influence in the estimated network

| Node | Influence-1 | Influence-2 |
|------|-------------|-------------|
| PCL_Q11 | 1.17 | 2.32 |
| GAD_Q2 | 1.12 | 2.22 |
| PCL_Q4 | 1.09 | 2.18 |
| PHQ_Q2 | 1.08 | 2.10 |
| GAD_Q1 | 1.04 | 2.08 |
| PCL_Q9 | 1.04 | 1.99 |
| PCL_Q1 | 1.03 | 2.08 |
| PCL_Q18 | 1.03 | 1.94 |
| PCL_Q3 | 1.03 | 1.99 |
| PCL_Q13 | 1.01 | 1.95 |
| PHQ_Q6 | 1.01 | 1.99 |
| PHQ_Q1 | 1.00 | 1.96 |
| GAD_Q4 | 0.99 | 1.92 |
| PCL_Q7 | 0.99 | 1.95 |
| PCL_Q5 | 0.99 | 2.01 |
| PCL_Q6 | 0.97 | 1.87 |
| PHQ_Q7 | 0.97 | 1.85 |
| GAD_Q3 | 0.95 | 1.93 |
| PCL_Q10 | 0.93 | 1.87 |
| PCL_Q15 | 0.90 | 1.67 |
| PHQ_Q5 | 0.89 | 1.71 |
| GAD_Q7 | 0.88 | 1.76 |
| PCL_Q14 | 0.88 | 1.77 |
| PCL_Q2 | 0.88 | 1.68 |
| PHQ_Q4 | 0.85 | 1.65 |
| PCL_Q17 | 0.83 | 1.62 |
| PHQ_Q8 | 0.82 | 1.57 |
| GAD_Q6 | 0.79 | 1.50 |
| PCL_Q20 | 0.79 | 1.53 |
| PCL_Q16 | 0.70 | 1.31 |
| PHQ_Q9 | 0.64 | 1.24 |
| PCL_Q8 | 0.54 | 1.05 |

## Table 3 | Top three edges within each community and across communities

| From | To | Weight | 95%CI lower | 95%CI upper |
|------|------|--------|-------------|-------------|
| GAD_Q2 | GAD_Q3 | 0.458 | 0.355 | 0.56 |
| GAD_Q1 | GAD_Q2 | 0.313 | 0.221 | 0.404 |
| GAD_Q1 | GAD_Q4 | 0.233 | 0.13 | 0.335 |
| PCL_Q6 | PCL_Q7 | 0.476 | 0.383 | 0.569 |
| PCL_Q17 | PCL_Q18 | 0.352 | 0.238 | 0.465 |
| PCL_Q13 | PCL_Q14 | 0.283 | 0.186 | 0.379 |
| PHQ_Q2 | PHQ_Q6 | 0.299 | 0.21 | 0.388 |
| PHQ_Q1 | PHQ_Q2 | 0.28 | 0.18 | 0.381 |
| PHQ_Q7 | PHQ_Q8 | 0.273 | 0.173 | 0.372 |
| GAD_Q6 | PCL_Q15 | 0.279 | 0.18 | 0.377 |
| PHQ_Q9 | PCL_Q16 | 0.152 | 0.021 | 0.283 |
| PHQ_Q4 | GAD_Q4 | 0.138 | 0.049 | 0.226 |

contained all GAD-7 items except item 6. GAD-1, GAD-2, GAD-3, and GAD-7 showed perfect replication (1.0), while GAD-4 demonstrated replication of 0.998. Community 3 (Externalizing Behaviors) included PCL-15, GAD-6, and PCL-16 with moderate stability. Replication rates were 0.694 for PCL-15, 0.676 for GAD-6, and 0.636 for PCL-16. Community 4 (Intrusion and Avoidance) contained PCL items PCL-1 through PCL-7, with most showing replication rates above 0.93. PCL-8 showed lower stability (0.608) with cross-loadings to Community 5. Community 5 (Negative Alterations) comprised PCL-9, PCL-10, PCL-11, PCL-13, and PCL-14. Replication rates ranged from 0.688 (PCL-13) to 0.912 (PCL-9, PCL-10). Community 6 (Anxious Arousal) included PCL-17 and PCL-18, both showing replication rates of 0.504.

### Dimension stability
Bootstrap analysis across [please include here: number of bootstrap samples] replications revealed varying levels of item stability within communities (Fig. 4). Items with replication rates above 70% included all PHQ items (ranging from 95.0% to 99.4%), all GAD items except GAD-6 (99.8%–100%), and most PCL intrusion/avoidance items (60.8%–99.6%). Items with lower dimensional stability (below 70%) included PCL-20 (30.6%), PCL-17 and PCL-18 (both 50.4%), PCL-16 (63.6%), PCL-15 (69.4%), and GAD-6 (67.6%), as shown in Table 4.

### Discussion
Network analysis of mental health symptoms among 317 family members of veterans and first responders revealed six distinct symptom communities: depression, generalized anxiety, intrusion and avoidance, anxious arousal, externalizing behaviors, and negative alterations. The network demonstrated stability with a correlation coefficient of 0.70, and identified PCL-11 (strong negative emotions) and GAD-2 (uncontrollable worry) as the most central nodes.

Our results are consistent with previous community detection network studies that found intrusion and avoidance symptoms in the same community in first responder[27], veteran[26], and civilian samples[28]. This suggests that the fear-based posttraumatic stress disorder symptoms of trauma-exposed family members are consistent with veterans and first responders themselves. In a sample of treatment-seeking emergency medical technicians and firefighters, internal intrusions had direct effects on external intrusions and avoidance symptoms using an 8-factor model of the PCL-5[37]. Investigating posttraumatic stress symptoms among spouses of veterans of the Yom Kippur War using latent class growth analysis, researchers found four trajectories over 12 years, with most falling in the resilience group, followed by recovery, chronic increase, and delayed patterns[38]. Posttraumatic stress symptoms can reduce relationship quality of first responders, which can manifest as anger outbursts, snappiness, and grumpiness among

connections both within and between established conceptual clusters. Table 3 presents the strongest edges within each community and across communities. The strongest within-community edges included GAD-2 to GAD-3 (weight = 0.458, 95% CI [0.355, 0.56]), PCL-6 to PCL-7 (weight = 0.476, 95% CI [0.383, 0.569]), and PHQ-2 to PHQ-6 (weight = 0.299, 95% CI [0.21, 0.388]). Cross-community connections were observed, with the strongest being GAD-6 to PCL-15 (weight = 0.279, 95% CI [0.18, 0.377]).

### Community detection
Multidimensional scaling network graph grouped symptoms by theoretical constructs is presented in Fig. 2. Exploratory Graph Analysis presented in Fig. 3 identified a six-community solution as the most stable across bootstrap replications. The six communities were: (1) Depression, (2) Generalized Anxiety, (3) Externalizing Behaviors, (4) Intrusion and Avoidance, (5) Negative Alterations, and (6) Anxious Arousal.

Community 1 (Depression) comprised primarily PHQ-9 items with high stability. As shown in Table 4, most PHQ items demonstrated replication rates above 0.95, with PHQ-1, PHQ-4, and PHQ-5 showing replication rates of 0.994, 0.992, and 0.992, respectively. PCL-20 (sleep difficulties) showed variable assignment across communities with lower stability (0.306 in Community 1). Community 2 (Generalized Anxiety)

**Fig. 2 | Multidimensional scaling network graph grouped by theoretical constructs (*n* = 317).** Network visualization showing partial correlations between mental health symptoms after controlling all other associations. Nodes represent individual items from standardized measures: circles represent posttraumatic stress disorder symptoms (PCL-5), squares represent depression symptoms (PHQ-9), and triangles represent generalized anxiety symptoms (GAD-7). Edge thickness and darkness correspond to the strength of partial correlations between symptoms. Green edges indicate positive correlations, red edges indicate negative correlations. Node colors group symptoms by their original theoretical constructs: posttraumatic stress disorder symptoms (blue), depression symptoms (red), and generalized anxiety symptoms (green). Layout is based on multidimensional scaling to optimize node positioning for interpretability.

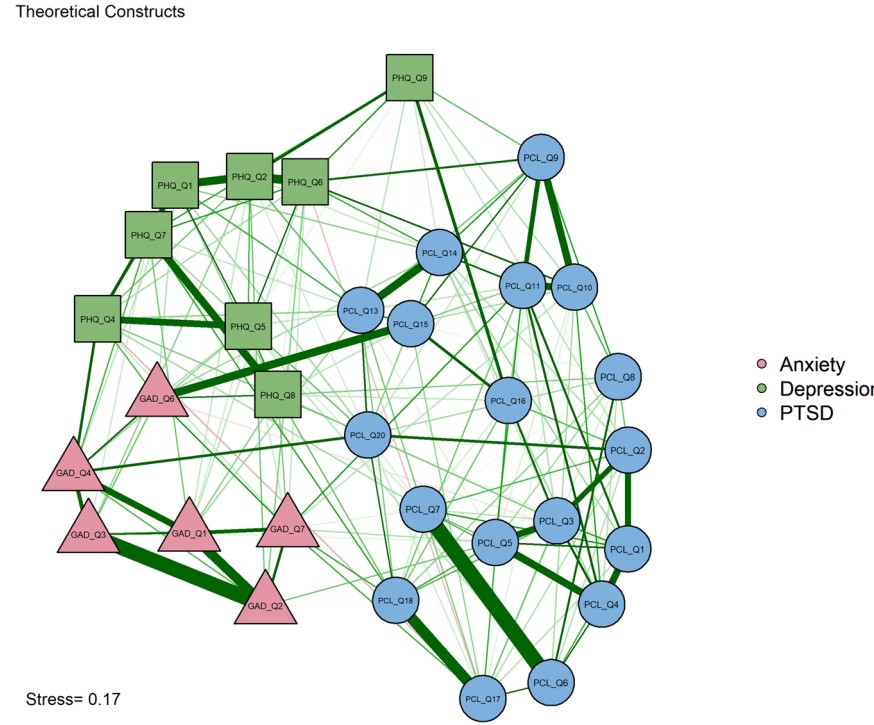

**Fig. 3 | Multidimensional scaling network graph grouped by empirical EGA communities (*n* = 317).** Network visualization showing the same partial correlation structure as Fig. 2a but with nodes colored according to the six communities identified through Exploratory Graph Analysis. Node shapes maintain the same meaning: circles (posttraumatic stress disorder symptoms), squares (depression symptoms), and triangles (generalized anxiety symptoms). Edge properties are identical to Fig. 2a. Node colors represent the six empirically-derived communities: Depression (light blue), Generalized Anxiety (light green), Externalizing Behaviors (pink), Intrusion & Avoidance (yellow), Negative Alterations (light purple), and Anxious Arousal (light orange). This grouping reflects data-driven symptom clustering rather than theoretical diagnostic categories.

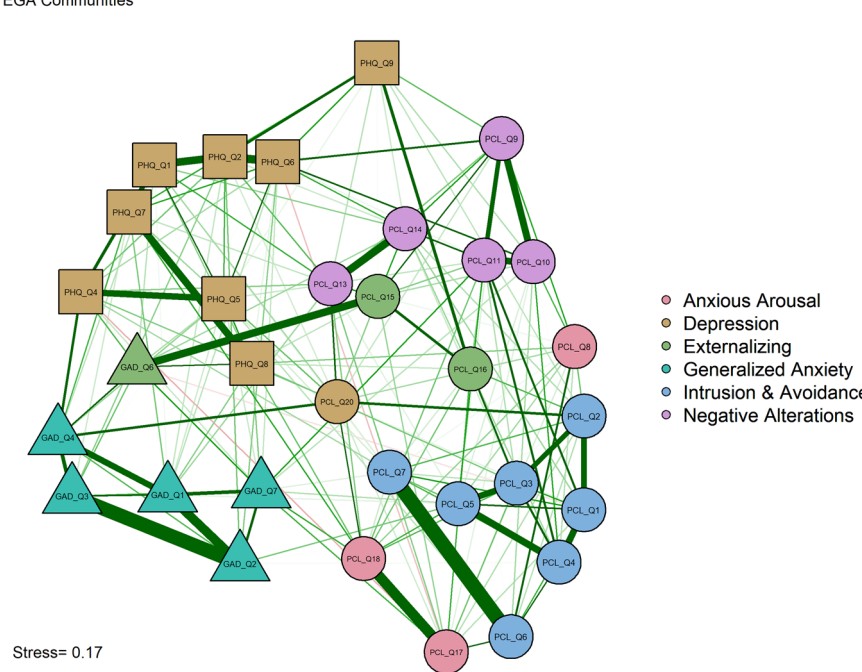

first responders and veterans alike[39]. Among military spouses, access to resources and psychoeducation about posttraumatic stress disorder symptoms can increase the capability to be resilient[40].

Our analyses found that major depressive disorder symptoms and generalized anxiety disorder symptoms formed separate communities, which is consistent with samples of veterans[26] and first responders[27]. However, in a civilian sample, major depressive disorder symptoms and generalized anxiety disorder symptoms formed their own community[28]. In a three-wave longitudinal study of National Guard spouses, researchers found that the only statistically significant predictor of post-deployment depression was pre-deployment depression[41]. The differing factor structure of the

PHQ-9 is important because among combat veterans[4] and first responders[42], the cognitive affective factor mediates posttraumatic stress disorder to suicide, whereas somatic depression does not. Given that major depressive disorder and generalized anxiety disorder formed their own community, further research is needed to understand these connections.

Anxious arousal formed its own community, which is consistent with previous research[28]. In a first responder sample, dysphoric arousal (including sleep disturbance) was included in the arousal and sleep community[27]. Among first responders, anxious arousal is adaptive regardless of first responder sub-type[9]. Divergent from previous studies, externalization and irritability formed their own community. Among first

**Table 4 | Item-replications within EGA communities**

|        | 1     | 2     | 3     | 4     | 5     | 6     | 7     |
|--------|-------|-------|-------|-------|-------|-------|-------|
| PHQ_Q1 | 0.994 | 0     | 0     | 0     | 0.002 | 0.004 | 0     |
| PHQ_Q4 | 0.992 | 0.004 | 0     | 0     | 0     | 0.004 | 0     |
| PHQ_Q5 | 0.992 | 0.004 | 0     | 0     | 0     | 0.004 | 0     |
| PHQ_Q7 | 0.978 | 0.002 | 0.008 | 0     | 0.002 | 0.01  | 0     |
| PHQ_Q8 | 0.976 | 0.002 | 0.01  | 0     | 0.002 | 0.01  | 0     |
| PHQ_Q2 | 0.952 | 0     | 0     | 0     | 0.03  | 0.002 | 0.016 |
| PHQ_Q6 | 0.95  | 0     | 0     | 0     | 0.032 | 0.002 | 0.016 |
| PHQ_Q9 | 0.95  | 0     | 0     | 0     | 0.032 | 0.002 | 0.016 |
| **PCL_Q20** | **0.306** | **0.024** | **0.136** | **0.208** | **0.112** | **0.206** | **0.008** |
| GAD_Q1 | 0     | 1     | 0     | 0     | 0     | 0     | 0     |
| GAD_Q2 | 0     | 1     | 0     | 0     | 0     | 0     | 0     |
| GAD_Q3 | 0     | 1     | 0     | 0     | 0     | 0     | 0     |
| GAD_Q7 | 0     | 1     | 0     | 0     | 0     | 0     | 0     |
| GAD_Q4 | 0.002 | 0.998 | 0     | 0     | 0     | 0     | 0     |
| **PCL_Q15** | **0.006** | **0.01**  | **0.694** | **0** | **0.282** | **0.006** | **0.002** |
| **GAD_Q6**  | **0.008** | **0.058** | **0.676** | **0** | **0.25**  | **0.006** | **0.002** |
| **PCL_Q16** | **0.048** | **0**     | **0.636** | **0.054** | **0.228** | **0.022** | **0.012** |
| PCL_Q1 | 0     | 0     | 0     | 0.996 | 0.004 | 0     | 0     |
| PCL_Q2 | 0     | 0     | 0     | 0.996 | 0.004 | 0     | 0     |
| PCL_Q3 | 0     | 0     | 0     | 0.996 | 0.004 | 0     | 0     |
| PCL_Q4 | 0     | 0     | 0     | 0.996 | 0.004 | 0     | 0     |
| PCL_Q5 | 0     | 0     | 0     | 0.996 | 0.004 | 0     | 0     |
| PCL_Q6 | 0     | 0     | 0     | 0.936 | 0.016 | 0.044 | 0.004 |
| PCL_Q7 | 0     | 0     | 0     | 0.936 | 0.016 | 0.044 | 0.004 |
| **PCL_Q8**  | **0.002** | **0** | **0.014** | **0.608** | **0.312** | **0.064** | **0** |
| PCL_Q9  | 0 | 0 | 0.05  | 0.032 | 0.912 | 0 | 0.006 |
| PCL_Q10 | 0 | 0 | 0.05  | 0.032 | 0.912 | 0 | 0.006 |
| PCL_Q11 | 0 | 0 | 0.05  | 0.034 | 0.91  | 0 | 0.006 |
| **PCL_Q14** | **0.07** | **0** | **0.18**  | **0** | **0.694** | **0.046** | **0.01** |
| **PCL_Q13** | **0.07** | **0** | **0.182** | **0** | **0.688** | **0.05**  | **0.01** |
| **PCL_Q17** | **0**    | **0** | **0.224** | **0.232** | **0.036** | **0.504** | **0.004** |
| **PCL_Q18** | **0**    | **0** | **0.224** | **0.232** | **0.036** | **0.504** | **0.004** |

Items below 0.70 replication within a single dimension are bold.

responders, externalizing behaviors are associated with emotional detachment, which also has a maladaptive impact on first responder children[43]. Among Vietnam War veterans, arousal negatively predicts marital satisfaction[44]. Comparing veteran and nonveteran transgenerational factors associated with military service, researchers found that veteran children were more likely to have externalized behavioral conditions as opposed to nonveterans[18]. Externalizing behaviors are associated with problematic alcohol use among veterans[45].

Negative affect symptoms and anhedonia symptoms formed their own community, called Negative Alterations. Previous research has found that these two constructs underlie this DSM-5 posttraumatic stress disorder symptom cluster[46]. These clusters and their associated negative emotions predict longitudinal linkage of psychopathology[47]. Anhedonia symptoms (decreased interest, detachment from others, and inability to experience positive emotions) are problematic in first responder research[48]. Anhedonia is related to poorer brain structural connectivity after trauma exposure[49].

### Theoretical context
Due to the heterogeneity of posttraumatic stress disorder, there have recently been theoretical advancements in the field of traumatic stress reactions[50]. The middle-out approach combines transdiagnostic and multidimensional frameworks in conjunction with a person-centered approach to conceptualize traumatic stress reactions[50]. Our separate anxiety and depression communities are consistent with the higher-order construction of these disorders. The middle-out lower-order items suggest the Negative Alterations community is consistent with the DSM-5 posttraumatic stress disorder symptom cluster. Using a nationally representative sample of veterans, researchers using latent class analysis investigating the interrelatedness of posttraumatic stress disorder, generalized anxiety, and depression found a five-class solution: low traumatic reaction, avoidant arousal, anxious/depressive, dysphoric arousal, and high traumatic stress reaction[50].

Our six symptom communities align with the middle-out approach's emphasis on dimensional presentations that transcend diagnostic boundaries. Within this framework, our central nodes (PCL-11 strong negative emotions and GAD-2 uncontrollable worry) represent transdiagnostic middle-order processes that connect across diagnostic categories. The identification of an externalizing behaviors community not prominent in previous studies highlights the framework's value in capturing contextualized symptom expressions unique to different trauma-exposed populations.

Veterans seeking mental health treatment who experience military sexual trauma or combat exposure had more severe depression and posttraumatic stress disorder symptoms compared to veterans not having those experiences[51]. Our study found that the PCL-5 blame item was in the Negative Alterations community. This is important because veterans who endorse blame have significantly higher depression and posttraumatic stress disorder[52]. In a treatment-seeking sample of active-duty veterans who were randomly assigned to group or individual cognitive processing therapy to examine the co-occurrence of posttraumatic stress disorder and depression, researchers found that posttraumatic stress disorder DSM-5 clusters of negative alterations in cognitions and mood, alterations in arousal and reactivity, and general distress factor predicted depression at baseline and posttreatment[53].

In network analysis examining the centrality and bridge symptoms in a U.K. veteran treatment-seeking sample, researchers found that emotional dysregulation, interpersonal difficulties, and negative self-concept were the strongest bridge symptoms between moral injury and posttraumatic stress disorder[54]. In a separate network analysis using a UK treatment-seeking veteran sample investigating the link between posttraumatic stress disorder and functional impairment, researchers found that diminished interest had the highest bridge strength[55]. In a longitudinal study of veterans over three years using network analysis, intrusion and avoidance item-level questions were most central to the network[56]. They also found that negative emotions, which are in our Negative Alterations community, were central to the network[56].

### Limitations
This study has several limitations. Most participants were married (58%), white (71%), and female (82%). The sample was well educated as 65% had some college or greater. One limitation is their familial relationship status. Over half of the sample was either the significant other of a veteran or first responder. Also, under the family category, 40% of the sample met those criteria. This could include either immediate and/or extended family relationships (sibling, aunt, uncle, cousin, grandparent). However, given the average age of this sample, we believe it to be immediate family as opposed to grandparents. An additional limitation is our reliance on self-report questionnaires, which may introduce response biases and limit the objectivity of our findings. Additionally, we did not include measures of resilience. Future studies could benefit from analyzing networks split by gender or familial relationship type, which would provide more nuanced insights into how these demographic factors might influence the observed outcomes.

An additional consideration is the distinction between intergenerational trauma and vicarious trauma in our sample. While we referenced intergenerational trauma literature in our Introduction, our sample composition suggests that vicarious trauma may be more applicable for many participants. With only 2.2% adult children in our sample and the majority

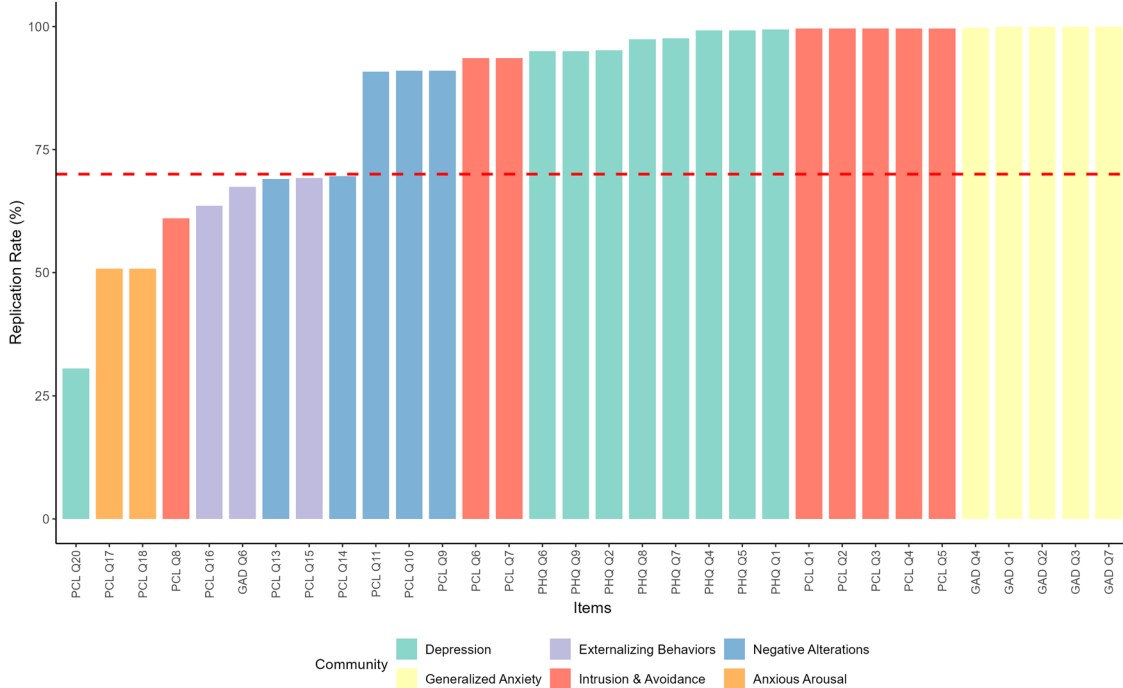

**Fig. 4 | Dimension stability of the empirical EGA solution across bootstraps.**
Bars show item replication rates within each of the six communities identified by
Exploratory Graph Analysis across bootstrap samples ($n = 317$ participants). Each
bar represents the percentage of bootstrap samples in which an item was assigned
to its most typical community. Colors represent different communities:

Depression (light blue), Generalized Anxiety (light green), Externalizing Beha-
viors (pink), Intrusion & Avoidance (yellow), Negative Alterations (light purple),
and Anxious Arousal (light orange). The horizontal red dashed line at 70% indi-
cates the threshold for adequate dimensional stability. Items above this threshold
demonstrate consistent community assignment across bootstrap resamples.

being spouses (57.1%) and extended family members (40.7%), the inter-
generational trauma findings from the literature may not fully apply to this
specific sample composition. Our findings may be more relevant to
understanding vicarious trauma processes in family members who are
primarily spouses and extended family rather than offspring experiencing
intergenerational transmission of trauma.

## Conclusion
This network analysis of mental health symptoms among family members
of veterans and first responders reveals a complex web of interconnected
psychological challenges. The identification of six distinct symptom com-
munities, centered around depression, anxiety, trauma responses, and
behavioral changes, demonstrates that these family members face unique
mental health challenges.

## Data availability
The raw clinical data are from a treatment-seeking group of family members
who consented to clinical treatment and use in possible research studies.
However, they did not give permission for their data to be open-access. The
numerical data underlying all figures and tables in this manuscript are
publicly available at Zenodo: https://doi.org/10.5281/zenodo.15860729.
These processed data files contain the statistical results necessary to
reproduce all visualizations and tables without compromising participant
confidentiality.

## Code availability
All analysis code supporting the conclusions of this study is available at
Zenodo: https://doi.org/10.5281/zenodo.15860629. The code includes
complete R scripts for network estimation, community detection, bootstrap
stability analysis, and high-resolution figure generation.

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

## Acknowledgements
We would like to thank the families of our veterans and first responders who selflessly serve. The authors received no specific funding for this work.

## Author contributions

Johannes A. Karl: Study design, data analysis, manuscript writing. Warren N. Ponder: Study conceptualization, data collection, manuscript writing. Jose Carbajal: Study supervision, manuscript writing and reviewing. Oleg N. Medvedev: Study conceptualization, manuscript writing and reviewing

## Competing interests

The authors declare no competing interests.
