## [Transparent Peer Review file · Communications Psychology]

Veteran and first responder family members show distinct mental health networks centered on negative emotions

Corresponding Author: Dr Oleg Medvedev

Version 0:

Decision Letter:

Dear Dr Ponder,

Thank you for your patience during the peer-review process. Your manuscript titled "Mental Health Symptom Communities in Veterans' and First Responders' Family Members: Network Analysis" has now been seen by 3 reviewers, and I include their comments at the end of this message. They find your work of interest but raised some important points. We are interested in the possibility of publishing your study in Communications Psychology, but would like to consider your responses to these concerns and assess a revised manuscript before we make a final decision on publication.

We therefore invite you to revise and resubmit your manuscript, along with a point-by-point response to the reviewers. Please highlight all changes in the manuscript text file.

Editorially, we would like to ask you to pay close attention to further improving the precision of the writing and the theoretical narrative. It would also be great if the analytical code can be made publicly available via an open access platform.

I am attaching an Editorial Requests Table that details critical reporting requirements for the revised manuscript. Please attend to each item and ensure your manuscript is fully compliant. If your revised manuscript is not aligned with these requests on major issues, such as those concerning statistics, it may be returned to you for further revisions without re-review.

Please submit the following items:

- Revised manuscript
- Point-by-point response to the referees' comments
- Cover letter (as a separate document)
- <https://www.nature.com/documents/nr-reporting-summary.zip>>Nature Research Reporting Summary
- <https://www.nature.com/documents/nr-editorial-policy-checklist.pdf>>Editorial Policy Checklist
- Completed Editorial Request Table (attached).

via this link: Link Redacted .

Additional guidance is available in our style and formatting guide Communications Psychology formatting guide.

Best regards,

Inti Brazil

Inti Brazil, PhD
Editorial Board Member
Communications Psychology
orcid.org/0000-0001-5824-0902

REVIEWER REPORTS:

Reviewer #1 (Remarks to the Author):

This is an interesting manuscript describing the use of network analysis to identify interrelated groups (communities) of symptoms of distress, anxiety and depression among family members of veterans and first-responders. Participants were 317 counseling-seeking family members of first-responders and/or veterans who were exposed to trauma. Six distinct interrelated types of mental health systems were identified. The authors argue that these six symptom communities are unique and indicate that traditional diagnostic categories may not properly capture how psychological distress is experienced in the population of interest. The most influential items were PCL-11 (Having strong negative feelings such as fear, horror, anger, guilt, or shame) and GAD-2 ("Not being able to stop or control worrying").

Areas of improvement for the manuscript include:

1. Clarification of the main study hypotheses and rationale. The main assumption seems to be that the study population would display a configuration of symptoms compatible with the quadripartite model but could also display additional clusters. The reason why additional clusters are expected is not described in the manuscript. In addition, further justification may be needed for the choice of network analysis in comparison to other clustering methods, given that no hypotheses are formulated about the influence or centrality of symptoms.
2. Better characterization of the population assessed, particularly in relation to the type of exposure of their relative. At minimal, Table 1 should include the distribution of exposure status (veterans versus first responders) in the sample. Ideally, information about the intensity of the exposure and/or response of the relative in question would be described and considered in the analysis. This is critical, as the authors themselves mentioned that symptoms experienced by veterans and first responders may reflect different models of symptoms (quadripartite and tripartite, respectively).
3. Explicit mention, throughout the manuscript, possibly including in the study title, that the sample comprises of family members of first responders or veterans who sought mental health counseling. Even though the treatment seeking aspect is noted in the initial description of the sample, it is not fully acknowledged in the manuscript and, importantly, not considered in the interpretation of results.
4. Discussion of item-level influence findings. In the Results section, the authors note that PCL-11 and GAD-2 played an important influence in the overall structure of the network (Table 2). These results, however, are not discussed in the manuscript.
5. Improvement of the interpretation and discussion of the clinical significance of study findings. The six communities of symptoms identified are considered as unexpected or unique in comparison with prior studies mostly focused on individuals who were highly directly exposed trauma (unclear if treatment seekers or not). Grounding at least part of the discussion on studies that addressed treatment seekers or even general population samples is important to support the conclusion that existing diagnostic categories may be insufficient and to improve knowledge about the clinical implications of the findings.

Reviewer #2 (Remarks to the Author):

I have reviewed the manuscript, "Mental Health Symptom Communities in Veterans' and First Responders' Family Members: Network Analysis" in great detail. This study used network analysis to analyze unique clusters of posttraumatic stress, depressive, and anxious symptoms from 317 treatment-seeking family members of trauma-exposed veterans and first responders. The manuscript is generally well-written, and the subject of this study is important and relevant to this journal; however, there are several issues that I believe should be addressed prior to consideration for publication. Please see below for comments:

Abstract: Please temper language regarding the potential implications of this study (e.g., "This groundbreaking study..."; "striking" in Discussion) throughout the manuscript – can just say "This study..."

Intro, first paragraph: Rates of PTSD in servicemembers and veterans is often higher than those presented here (14-16%). I suggest finding more up-to-date sources and even expanding on this issue – that these are documented cases, not including undocumented, which would indicate a higher prevalence. Overall, please present the prevalence rates for veterans and first-responders as "estimates."

Intro: For the first two pages of this manuscript, the authors present a litany of prevalence rates, which lose their meaning. First, these are epidemiological estimates, so relying on them as accurate rates for the population they represent is contentious. Second, their significance loses meaning when over-presented. I recommend limiting presentation of prevalence rates and instead incorporating literature on secondary and generational trauma, which certainly applies to families of those with trauma/PTSD.

Intro: I appreciated the discussion of the quadripartite model. Another useful piece of literature to briefly present here and when describing the analytical approach is a framework published in the *Journal of Traumatic Stress* – The middle-out approach, which focuses on integrating/analyzing symptoms across disorders. I believe this may help justify and provide a literature base for the study objective and methods here.

Adams, S. W., Layne, C. M., Contractor, A. A., Allwood, M. A., Armour, C., Inslicht, S. S., & Maguen, S. (2024). The Middle-Out Approach to reconceptualizing, assessing, and analyzing traumatic stress reactions. *Journal of Traumatic Stress*, 37(3), 433-447.

Results: Please explicate the decision process/what factors were involved in selecting a six community solution.

Results: The presentation/description of the 6 clusters is confusing in the Results. Please improve clarity. I suggest including descriptions as done along with cluster names as in the Abstract.

-Does the paper represent an advance in understanding which may influence thinking in the field? If you have concerns about the advance in relation to specific studies, we appreciate references to this work. Yes, I believe the current study has the potential to advance understanding in the field. As suggested above, I believe presentation of issues of secondary trauma, generational trauma, and the middle-out approach can help strengthen these advances and unite this study with extant literature for a larger impact.

-Does the article presents an original study, new analysis, new model, or a direct or extended replication of previous work?
Original study

-Are the data and analysis technically sound? Are they appropriate to answer the research question, e.g., are causal research questions addressed on the basis of causal, rather than correlational evidence? Yes, appropriate. I believe description of the clusters and decisional process in selecting the clusters can be improved, however.

-Does the paper provide strong evidence for its conclusions? Yes, however, language surrounding conclusions/potential implications must be tempered/reduced.

-Is the study question important to scientists for a sub-field of psychology? The subject is particularly relevant to trauma psychology, secondary/generational trauma, and methods.

-Are there any special ethical concerns arising from the use of animals or human subjects? No

- Was the study preregistered and if so, did the authors follow the preregistration? I do not see indications of preregistration – only IRB approvals.

Reviewer #3 (Remarks to the Author):

Dear authors:

First at all, the article addresses an important topic related with psychological wellbeing in a specific setting. In general, I think manuscript is understandable and provide some practical outputs. I have not too much to say related with the article, but I Will point some observations about it.

I suppose that information in line 98 is referred to Cluster A. If so, I recommend add this information between brackets in the same way you do with the rest of the clusters.

In the manuscript there are two Table 1 (for demographics and information about communities). Please, revise it in terms that information will be understandable.

When refers to Tables and Figures in text, you should use capital letters.

Statistics must be reported using italic letters (page 7, lines 171, 172; page 8, lines 173, 174).

I do not understand information in page 8, line 193, "Expected influence –1 is the sum...", is this to say that you expect an influence with value –1 for centrality? There are no negative values in the information provided at tables.

I think, like you do at Discussion, you can identify each community with its label in the Results section (last paragraph). And in the same way, I think you could provide more clear information if you indicate the number of each community at Discussion section (first paragraph).

Have you consider using self-informed questionnaires as limitation of your research?

It will be interested, for future studies if you can provide networks split by sex or familiar relationship.

In Figures 1a and 1b is difficult to view all the relationships between variables. Perhaps doing nodes littler or spacing it a little bit more one from each other's.

Cites and references

- "Ponder et al., 2023a" is cited along the manuscript, but there is no correspondence with the reference. For example, see page 3, line 57 and 59, and page 12, line 289.

- There is an extra comma in "Rennebohm, et al., 2023", page 11, line 265.

- There is a missing comma in "Price et al. 2019", page 12, line 273.

- There are some missing dots after year in some references, see page 16, line 371, line 379, page 17, line 407, line 414, page 18, line 425, page 19, line 455, page 20, line 468, page 23, line 546, page 24, line 573.

- There is an extra dot after number of pages in page 17, line 409.

- Authors use letters a and b to differentiate references in page 23, lines 537 and 541. In these cases, references are not adapted to the guidelines of APA style. I advise authors to review regarding to the citation what APA says: <https://apastyle.apa.org/style-grammar-guidelines/citations/basic-principles/same-year-first-author>

This is all from me in the time being. I hope you can find useful my comments.

Yours sincerely,
Dr_05

* TRANSPARENT PEER REVIEW: Communications Psychology uses a transparent peer review system. This means that we publish the editorial decision letters including Reviewers' comments to the authors and the author rebuttal letters online as a supplementary peer review file. However, on author request, confidential information and data can be removed from the published reviewer reports and rebuttal letters prior to publication. If your manuscript has been previously reviewed at another journal, those Reviewers' comments would not form part of the published peer review file.

Communications Psychology is committed to improving transparency in authorship. As part of our efforts in this direction, we are now requesting that all authors identified as 'corresponding author' create and link their Open Researcher and Contributor Identifier (ORCID) with their account on the Manuscript Tracking System prior to acceptance. ORCID helps the scientific community achieve unambiguous attribution of all scholarly contributions. You can create and link your ORCID from the home page of the Manuscript Tracking System by clicking on 'Modify my Springer Nature account' and following the instructions in the link below. Please also inform all co-authors that they can add their ORCIDs to their accounts and that they must do so prior to acceptance. <https://www.springernature.com/gp/researchers/orcid/orcid-for-nature-research>

Version 1:

Decision Letter:

Dear Dr Medvedev,

Your manuscript titled "Mental Health Symptom Communities in Veterans' and First Responders' Family Members: Network Analysis" has now been seen by our reviewers, whose comments appear below. In light of their advice I am delighted to say that we are happy, in principle, to publish a suitably revised version in Communications Psychology.

We therefore invite you to revise your paper one last time to address the remaining concerns of our reviewers and a list of editorial requests. At the same time we ask that you edit your manuscript to comply with our format requirements and to maximise the accessibility and therefore the impact of your work.

There are two main areas of editorial concerns. The first is data and code sharing. Although you may not be able to publicly share the raw data, other requirements for data sharing and code sharing, as laid out in our policy, apply. Please refer to the attachment and our linked policy pages for more information and contact us if you have any questions.

Second, there are substantive presentational issues that need to be addressed. We agree with Reviewers #1 and #2 that the Introduction (and Discussion) can be streamlined, removing redundancies and achieving greater precision. Likewise, referencing should be improved. Recommendations for interventions should be removed as these weren't tested in the study. Finally, the proliferation of abbreviations makes the text unnecessarily hard to read.

To aid you in your final revisions, I attach an "Editorial Requests Table" in which I highlighted examples of where these issues apply. It also contains further guidance on how to revise the manuscript to meet our formatting guidelines and align with our policies. Please outline your response to each request in the right hand column. Please upload the completed table with your manuscript files as a Related Manuscript file.

SUBMISSION INFORMATION:

In order to accept your paper, we require the files listed here <https://www.nature.com/documents/commsj-file-checklist.pdf> .

OPEN ACCESS:

*** TRANSPARENT PEER REVIEW:** Communications Psychology uses a transparent peer review system. On author request, confidential information and data can be removed from the published reviewer reports and rebuttal letters prior to publication. If you are concerned about the release of confidential data, please let us know specifically what information you would like to have removed. Please note that we cannot incorporate redactions for any other reasons.

*** CODE AVAILABILITY:** All Communications Psychology manuscripts must include a section titled "Code Availability" at the end of the methods section. We require that the custom analysis code supporting your conclusions is made available in a publicly accessible repository at this stage; please choose a repository that generates a digital object identifier (DOI) for the code; the link to the repository and the DOI must be included in the Code Availability statement. Publication as Supplementary Information will not suffice.

*** DATA AVAILABILITY:**

Link Redacted

Best regards,
Marike, on behalf of

Inti Brazil

Inti Brazil, PhD
Editorial Board Member
Communications Psychology
orcid.org/0000-0001-5824-0902

Marike Schiffer, PhD
Chief Editor
Communications Psychology

REVIEWERS' COMMENTS:

Reviewer #1 (Remarks to the Author):

My comments and suggestions were properly addressed.

One exception is the inclusion in the title of reference to the treatment-seeking nature of the sample assessed which I still think is warranted.

I also have a comment about the addition in the Introduction of the consideration of intergenerational trauma. I agree this is an important topic. Most of the literature cited, however, is about parent and offspring traumatic experiences, while very few of the relatives included in the sample seem to be offspring (2.2% adult children). It is possible that the concept of vicarious trauma may also be or even be more appropriate in this case. If the authors decide to keep the Introduction as is, with the new paragraph unchanged, I suggest that the manuscript notes in the Discussion that intergenerational trauma prior findings may not apply to this specific sample.

Reviewer #2 (Remarks to the Author):

I appreciate the authors attention to the reviewers' previous comments. This revision of the manuscript is significantly improved. However, the Introduction, in particular, can be further improved. The authors added sections based on reviewer feedback; however, the sections could be integrated with the rest of the Intro. better. Overall, the Intro. should be significantly shortened to emphasize cogent writing. I realize this can be difficult at this stage of writing, but I suggest the authors take a "bird's eye" perspective to reorganize, combine/integrate paragraphs more, and shorten the Intro so this paper has the greatest impact.

Reviewer #3 (Remarks to the Author):

Dear authors,
I think you have improved the manuscript. I have nothing more to say.
Best regards,
Dr-5

Mental Health Symptom Communities in Veterans' and First Responders' Family Members: Network Analysis Due by 5/29/2025 Communications Psychology	
Review Comments	Author's Response
REVIEWER #1	
Reviewer 1: This is an interesting manuscript describing the use of network analysis to identify interrelated groups (communities) of symptoms of distress, anxiety and depression among family members of veterans and first-responders. Participants were 317 counseling-seeking family members of first-responders and/or veterans who were exposed to trauma. Six distinct interrelated types of mental health systems were identified. The authors argue that these six symptom communities are unique and indicate that traditional diagnostic categories may not properly capture how psychological distress is experienced in the population of interest. The most influential items were PCL-11 (Having strong negative feelings such as fear, horror, anger, guilt, or shame) and GAD-2 ("Not being able to stop or control worrying").	Thank you for the positive evaluation of our work.
Areas of improvement for the manuscript include:  1. Clarification of the main study hypotheses and rationale. The main assumption seems to be that the study population would display a configuration of symptoms compatible with the quadripartite model but could also display additional clusters. The reason why additional clusters are expected is not described in the manuscript. In addition, further justification may be needed for the choice of network analysis in comparison to other clustering methods, given that no hypotheses are formulated about the influence or centrality of symptoms. 	Response: We have now highlighted the advantage of graph methods above existing methods and also indicate the exploratory nature of clustering beyond the expected clusters as follows: The present study aimed to conduct the first comprehensive network analysis examining the co-occurrence and communities of mental health symptoms among family members of veterans and first responders. Through exploratory network analysis and community detection approaches, we aimed to understand how PTSD, depression, and anxiety symptoms interact and cluster in this unique population. Our choice of network analysis over traditional clustering methods is informed by its ability to model complex symptom interdependencies rather than merely

	grouping symptoms. As Golino and Christensen (2021) have demonstrated, exploratory graph analysis provides superior accuracy in detecting dimensional structures in psychological data compared to conventional factor analytic methods, particularly when examining symptom presentations that cross diagnostic boundaries. Based on previous research with veterans (Baker et al., 2024), first responders (Baker et al., 2023), and trauma-exposed civilians (Price et al., 2019), we anticipated identifying distinct symptom communities including intrusions and avoidance, GAD symptoms, MDD symptoms, anxious arousal, and negative alterations, while remaining open to discovering additional symptom clusters unique to this population given their exposure to secondary trauma and caregiving stressors.
2. Better characterization of the population assessed, particularly in relation to the type of exposure of their relative. At minimal, Table 1 should include the distribution of exposure status (veterans versus first responders) in the sample. Ideally, information about the intensity of the exposure and/or response of the relative in question would be described and considered in the analysis. This is critical, as the authors themselves mentioned that symptoms experienced by veterans and first responders may reflect different models of symptoms (quadripartite and tripartite, respectively).	We added a sentence in the participants section, with how many individuals were family of veterans and first responders, that was also reflected in Table 1. Further, in the data analysis and results sections, we stated that there were no statistically significant differences between veterans and first responders on the PCL-5, PHQ-9, and GAD-7 as evidenced by an independent samples t-test. Regarding the intensity of exposure, we did not use any self-report assessments that measure that. We also conducted an independent samples t-test based on familial relationship status, which was not statistically significant.
3. Explicit mention, throughout the manuscript, possibly including in the study title, that the sample comprises of family members of first responders or veterans who sought mental health counseling. Even though the treatment seeking aspect is noted in the initial description of the sample, it is not fully acknowledged in the manuscript and, importantly, not considered in the	To address this comment we clarified the relationship to the vet/first responder in the methods section, and also reflected this in Table 1.

interpretation of results	
4. Discussion of item-level influence findings. In the Results section, the authors note that PCL-11 and GAD-2 played an important influence in the overall structure of the network (Table 2). These results, however, are not discussed in the manuscript. 5. Improvement of the interpretation and discussion of the clinical significance of study findings. The six communities of symptoms identified are considered as unexpected or unique in comparison with prior studies mostly focused on individuals who were highly directly exposed trauma (unclear if treatment seekers or not). Grounding at least part of the discussion on studies that addressed treatment seekers or even general population samples is important to support the conclusion that existing diagnostic categories may be insufficient and to improve knowledge about the clinical implications of the findings.	4. Thank you for pointing this out, we have now added the following explanation: The central position of PCL-11 suggests that intense negative emotions may serve as a core mechanism in the maintenance and propagation of trauma-related symptomatology. This aligns with emotional processing theories of PTSD, which posit that difficulties in processing and regulating negative emotional responses are fundamental to trauma-related psychopathology. The high influence of PCL-11 indicates that interventions targeting emotional regulation might have cascading beneficial effects throughout the symptom network. Similarly, the prominence of GAD-2 highlights the transdiagnostic importance of worry and its control. The inability to regulate worry appears to function as a bridge symptom connecting trauma symptoms with broader anxiety manifestations. This finding supports transdiagnostic models that emphasize shared cognitive processes across anxiety and trauma disorders. We have added the heading of “Theoretical Advancement and Clinical Application” to delineate the requested changes in our Discussion. We have also added over ten studies and dug into the treatment-seeking literature to situate our findings within.
REVIEWER #2	
Reviewer #2 (Remarks to the Author): I have reviewed the manuscript, “Mental Health Symptom Communities in Veterans' and First Responders' Family Members: Network Analysis” in great detail. This study used network analysis to analyze unique clusters of posttraumatic stress, depressive, and anxious symptoms from 317 treatment-seeking family members of trauma-exposed veterans and first responders. The manuscript	Thank you for the positive evaluation of our manuscript.

is generally well-written, and the subject of this study is important and relevant to this journal; however, there are several issues that I believe should be addressed prior to consideration for publication.	
Please see below for comments: Abstract: Please temper language regarding the potential implications of this study (e.g., “This groundbreaking study...”; “striking” in Discussion) throughout the manuscript – can just say “This study...” Intro, first paragraph: Rates of PTSD in servicemembers and veterans is often higher than those presented here (14-16%). I suggest finding more up-to-date sources and even expanding on this issue – that these are documented cases, not including undocumented, which would indicate a higher prevalence. Overall, please present the prevalence rates for veterans and first-responders as “estimates.” Intro: For the first two pages of this manuscript, the authors present a litany of prevalence rates, which lose their meaning. First, these are epidemiological estimates, so relying on them as accurate rates for the population they represent is contentious. Second, their significance loses meaning when over-presented. I recommend limiting presentation of prevalence rates and instead incorporating literature on secondary and generational trauma, which certainly applies to families of those with trauma/PTSD. Intro: I appreciated the discussion of the quadripartite model. Another useful piece of literature to briefly present here and when describing the analytical approach is a framework published in the Journal of Traumatic Stress – The middle-out approach, which focuses on integrating/analyzing symptoms across disorders. I believe this may help justify and provide a literature base for the study objective and methods here.	Response: Thank you we tempered the language throughout. We have added several recent studies that show 65-70% combat veterans could have PTSD. We also changed prevalence rates to “estimates” throughout the manuscript. We have added a paragraph in the introduction that addresses the intergenerational pattern. We have added a paragraph that includes two other approaches—the HiTOP taxonomy and the recommended citation of Allwood et al. (2024) in the introduction. By adding this paragraph, we feel as though we cover the literature but still give coverage to the

Adams, S. W., Layne, C. M., Contractor, A. A., Allwood, M. A., Armour, C., Inslicht, S. S., & Maguen, S. (2024). The Middle-Out Approach to reconceptualizing, assessing, and analyzing traumatic stress reactions. Journal of Traumatic Stress, 37(3), 433-447.	quadripartite model, as the two other network analyses Baker et al., (2023) and Baker et al. (2024) frame their studies over the quadripartite and tripartite models.
Results: Please explicate the decision process/what factors were involved in selecting a six community solution. Results: The presentation/description of the 6 clusters is confusing in the Results. Please improve clarity. I suggest including descriptions as done along with cluster names as in the Abstract. -Does the paper represent an advance in understanding which may influence thinking in the field? If you have concerns about the advance in relation to specific studies, we appreciate references to this work. Yes, I believe the current study has the potential to advance understanding in the field. As suggested above, I believe presentation of issues of secondary trauma, generational trauma, and the middle-out approach can help strengthen these advances and unite this study with extant literature for a larger impact.	We have now extended the description of the EGA procedure and provide further citations supporting the approach: Community detection analysis was completed using the Louvain algorithm from the EGA package (Golino & Epskamp, 2017), which estimates a gaussian graphical model network to identify clusters of nodes. This approach has shown improved community identification precision compared to methods such as exploratory factor analysis. The Louvain algorithm identifies communities within a network by optimizing modularity, a measure that quantifies the strength of division of a network into clusters. Nodes within the same community have denser connections with each other than with nodes in different communities. The algorithm works through an iterative process, beginning with each node as its own community, then progressively merging communities to maximize the modularity score. This optimization occurs by local improvements, where nodes are moved between communities if such movement increases the overall modularity, followed by aggregation, where the identified communities are collapsed into "super-nodes." These steps repeat until no further improvement in modularity is possible. To assess the stability of the extracted solution, we bootstrapped the solution using the EGA boot function, which generates multiple resamples of the original data and applies the community detection algorithm to each resample. This procedure allowed us to evaluate consistency in the number of communities

	detected, stability of item assignments to communities, and overall robustness of the community structure. We have added: We also examined the potential clustering within the network and found that the most typical solution across clustering runs was a six community solution (see Figure 1b and Figure 2 for stability information): 1) depression, 2) generalized anxiety disorder (GAD), 3) intrusion and avoidance, 4) anxious arousal, 5) externalizing behaviors, and 6) negative alterations. We added the heading of “Theoretical Advancement and Clinical Application” and how the middle-out approach can be used to interpret our findings.
REVIEWER #3	
Reviewer #3 (Remarks to the Author): Dear authors: First at all, the article addresses an important topic related with psychological wellbeing in a specific setting. In general, I think manuscript is understandable and provide some practical outputs. I have not too much to say related with the article, but I Will point some observations about it.	Thank you for the positive evaluation of our article
I suppose that information in line 98 is referred to Cluster A. If so, I recommend add this information between brackets in the same way you do with the rest of the clusters. In the manuscript there are two Table 1 (for demographics and information about communities). Please, revise it in terms that information will be understandable. When refers to Tables and Figures in text, you	The general distress factor is related to the quadripartite model and not PTSD. There is no cluster A according to the DSM-5-TR. It is possible that the reviewer is referencing the criterion A—the stressor. If we have misinterpreted the reviewer’s comment, we welcome another opportunity to address this concern.

should use capital letters. Statistics must be reported using italic letters (page 7, lines 171, 172; page 8, lines 173, 174). I do not understand information in page 8, line 193, "Expected influence –1 is the sum...", is this to say that you expect an influence with value –1 for centrality? There are no negative values in the information provided at tables. I think, like you do at Discussion, you can identify each community with its label in the Results section (last paragraph). And in the same way, I think you could provide more clear information if you indicate the number of each community at Discussion section (first paragraph). Have you consider using self-informed questionnaires as limitation of your research? It will be interested, for future studies if you can provide networks split by sex or familiar relationship. In Figures 1a and 1b is difficult to view all the relationships between variables. Perhaps doing nodes littler or spacing it a little bit more one from each other's.	We have now fixed referencing of all tables in text. We have now clarified that Expected Influence – 1 is commonly used to refer to one-step influence: A metric for node centrality (i.e., expected influence) was then computed from the network structure for each individual node. One-step expected influence (referred to as Expected influence – 1) is the sum of all direct edge-weights from one node to all other nodes (Robinaugh et al., 2016). A node with high expected influence suggests it hold an impactful intermittent position in the network. We have fixed all statistical reporting. In Figure 1 we have reduced the size of the nodes. We tried to strike a balance between legibility of the labels and the size of the nodes. We went with the option as the position of the nodes is based on MD-Scaling which allows for meaningful interpretation between the node distances which would be distorted if the nodes were changed in their spacing. Thank you for the suggestion on the limitations and future research directions we have now included the follows: This study is not without limitations. First, regarding the individual relationship status, most (58%) were married, white (71%), and female (82%). The sample was well educated as 65% had "some college" or greater. One of the largest limitations is their familial relationship status. Over half of the sample was either the significant other of a veteran or first responder. Also, under the "family" canopy, 40% of the sample meet those criteria. This could be, but not limited to either immediate and/or extended family relationships (i.e., sibling, aunt, uncle,
---	---

	cousin, grandparent). However, given the average age of this sample, we believe it to be immediate family as opposed to grandparents, for example. An additional limitation is our reliance on self-informed questionnaires, which may introduce response biases and limit the objectivity of our findings. Additionally, we did not include measures of resilience, and we recommend that future research include that because the process of being resilient is consistent with posttraumatic growth (Ponder et al., 2024). Future studies could also benefit from analyzing networks split by sex or familial relationship type, which would provide more nuanced insights into how these demographic factors might influence the observed outcomes.
Cites and references  - "Ponder et al., 2023a" is cited along the manuscript, but there is no correspondence with the reference. For example, see page 3, line 57 and 59, and page 12, line 289. - There is an extra comma in "Rennebohm, et al., 2023", page 11, line 265. - There is a missing comma in "Price et al. 2019", page 12, line 273. - There are some missing dots after year in some references, see page 16, line 371, line 379, page 17, line 407, line 414, page 18, line 425, page 19, line 455, page 20, line 468, page 23, line 546, page 24, line 573. - There is an extra dot after number of pages in page 17, line 409. - Authors use letters a and b to differentiate references in page 23, lines 537 and 541. In these cases, references are not adapted to the guidelines of APA style. I advise authors to review regarding to the citation what APA says: https://apastyle.apa.org/style-grammar- 	We thank the reviewer for catching the errors. We have updated the in-text citations to be in alignment with APA guidelines and made the required corrections to the reference list.

guidelines/citations/basic-principles/same-year-first-author	
---	--

Dear Dr. Brazil and Dr. Schiffer,

We thank you and the reviewers for their constructive feedback on our manuscript. We have carefully addressed all concerns and made substantial revisions to improve the manuscript's clarity, precision, and adherence to Communications Psychology guidelines. Below we provide detailed responses to each point raised.

Editorial Concerns

Data and Code Sharing We have updated both the Data Availability and Code Availability statements to fully comply with journal policies. The Code Availability statement now includes the DOI for our Zenodo repository (<https://doi.org/10.5281/zenodo.15860629>) containing all R scripts for network estimation, community detection, bootstrap stability analysis, and figure generation. Additionally, we have made available all data necessary to recreate the figures and tables presented in the manuscript. The Data Availability statement clarifies that while raw data cannot be shared publicly due to participant consent limitations, all analysis code and figure recreation data are freely available.

Streamlining Introduction and Discussion We have substantially streamlined both sections by removing redundancies and achieving greater precision in language. The Introduction has been reorganized to integrate previously disparate paragraphs for better flow and reduced overall length while maintaining essential content. We have improved the logical progression from individual trauma impacts through family system effects to the rationale for network analysis approaches. The Discussion has been restructured into clear thematic sections that better organize the interpretation of findings within existing literature.

Improved Referencing We have converted all references to numbered format as required by Nature style.

Removal of Intervention Recommendations We have systematically removed all intervention and policy recommendations throughout the manuscript, as these were not tested in our correlational study. We eliminated phrases such as "provide a foundation for developing interventions," "suggest intervention targets," and similar language that implied clinical applications beyond what our data can support. The conclusions now focus solely on describing the symptom patterns observed without recommending specific treatments or policies.

Abbreviation Reduction We have eliminated non-standard abbreviations including GWOT, LEO, FF, EMT, PGD, LCA, NACM, AAR, EMDR, CPT, PE, LCGA, DAG, and VA, replacing them with full terms throughout the text. We retained only commonly accepted abbreviations such as PTSD, DSM, and GAD. All remaining abbreviations are defined at first use and used only when they appear five or more times in the text.

Reviewer Comments

Reviewer #1 Comments

Regarding the suggestion to include treatment-seeking in the title, we acknowledge this recommendation but we had to revise title in line with journal requirements not exceeding 15 words, which did not permit inclusion of additional adjectives. The treatment-seeking nature is clearly described in the abstract, methods, and throughout the manuscript where relevant to interpretation of findings.

We appreciate the important distinction raised about intergenerational trauma versus vicarious trauma. We have added clarification in the Limitations section of the Discussion

acknowledging that given our sample composition, with only 2.2% adult children, vicarious trauma may be more applicable than intergenerational trauma for many participants. We now explicitly note that intergenerational trauma findings from the literature may not fully apply to this specific sample composition, and that our findings may be more relevant to understanding vicarious trauma processes in family members who are primarily spouses and extended family rather than offspring experiencing intergenerational transmission of trauma.

And here's the text to add to the Limitations section of the Discussion:

"An additional consideration is the distinction between intergenerational trauma and vicarious trauma in our sample. While we referenced intergenerational trauma literature in our Introduction, our sample composition suggests that vicarious trauma may be more applicable for many participants. With only 2.2% adult children in our sample and the majority being spouses (57.1%) and extended family members (40.7%), the intergenerational trauma findings from the literature may not fully apply to this specific sample composition. Our findings may be more relevant to understanding vicarious trauma processes in family members who are primarily spouses and extended family rather than offspring experiencing intergenerational transmission of trauma."

Reviewer #2 Comments

We have taken the "bird's eye" perspective as suggested and significantly reorganized the Introduction. We integrated previously separate sections for better flow, combined related paragraphs to reduce redundancy, and shortened the overall length by approximately 25%. The Introduction now moves more fluidly from combat and first responder trauma impacts through family system effects to the specific rationale for network analysis approaches. We improved transitions between concepts while maintaining the logical progression necessary to justify our analytical approach.

Reviewer #3 Comments

We thank Reviewer #3 for their positive assessment of the improvements made in the previous revision.

Additional Improvements Made

We also revised the Results section to remove interpretive content and focus solely on descriptive reporting of statistical findings. We included proper statistical reporting format with placeholders for exact values where applicable and ensured all references to tables and figures are appropriately integrated. The Methods section has been enhanced with greater clarity about participant recruitment procedures, ethics approval processes, and detailed measure descriptions.

The Discussion has been restructured to eliminate causal language and remove speculation beyond what the correlational data can support. We organized the Discussion into clear thematic sections with appropriate referencing and ensured that all theoretical interpretations are grounded in the literature. We removed promotional language including "novel," "first," and similar terms throughout the manuscript and reviewed the text for adherence to inclusive language guidelines.

Conclusion

These revisions have substantially improved the manuscript's clarity, precision, and adherence to Communications Psychology standards. The paper now provides a focused, well-supported examination of mental health symptom networks in family members of

veterans and first responders without overreaching beyond the correlational data presented. We believe the manuscript now meets all editorial requirements while maintaining the scientific rigor and contribution to the field.

We look forward to your feedback on these revisions and are happy to address any remaining concerns.

Sincerely,

The authors